# Description of Three Novel Members in the Family *Geobacteraceae*, *Oryzomonas japonicum* gen. nov., sp. nov., *Oryzomonas sagensis* sp. nov., and *Oryzomonas ruber* sp. nov.

**DOI:** 10.3390/microorganisms8050634

**Published:** 2020-04-27

**Authors:** Zhenxing Xu, Yoko Masuda, Chie Hayakawa, Natsumi Ushijima, Keisuke Kawano, Yutaka Shiratori, Keishi Senoo, Hideomi Itoh

**Affiliations:** 1Department of Applied Biological Chemistry, Graduate School of Agricultural and Life Sciences, The University of Tokyo, Tokyo 113-8657, Japan; xuzx.ut@gmail.com (Z.X.); asenoo@mail.ecc.u-tokyo.ac.jp (K.S.); 2School of Agriculture, Utsunomiya University, Tochigi 321-8505, Japan; chayakawa@cc.utsunomiya-u.ac.jp; 3Support Section for Education and Research, Graduate School of Dental Medicine, Hokkaido University, Hokkaido 060-8586, Japan; ntm@den.hokudai.ac.jp; 4Department of Marine Biology and Sciences, School of Biological Sciences, Tokai University, Hokkaido 005-8601, Japan; kawano.1021@aist.go.jp; 5Niigata Agricultural Research Institute, Niigata 940-0826, Japan; sira@ari.pref.niigata.jp; 6Collaborative Research Institute for Innovative Microbiology, The University of Tokyo, Tokyo 113-8657, Japan; 7Bioproduction Research Institute, National Institute of Advanced Industrial Science and Technology (AIST) Hokkaido, Hokkaido 062-8517, Japan

**Keywords:** *Oryzomonas japonicum* sp. nov., *Oryzomonas sagensis* sp. nov., *Oryzomonas ruber* sp. nov., *Geobacteraceae*, paddy soil, pond sediment, iron-reducing bacteria

## Abstract

Bacteria of the family *Geobacteraceae* are particularly common and deeply involved in many biogeochemical processes in terrestrial and freshwater environments. As part of a study to understand biogeochemical cycling in freshwater sediments, three iron-reducing isolates, designated as Red96^T^, Red100^T^, and Red88^T^, were isolated from the soils of two paddy fields and pond sediment located in Japan. The cells were Gram-negative, strictly anaerobic, rod-shaped, motile, and red-pigmented on agar plates. Growth of these three strains was coupled to the reduction of Fe(III)-NTA, Fe(III) citrate, and ferrihydrite with malate, methanol, pyruvate, and various organic acids and sugars serving as alternate electron donors. Phylogenetic analysis based on the housekeeping genes (16S rRNA gene, *gyrB, rpoB, nifD, fusA,* and *recA*) and 92 concatenated core genes indicated that all the isolates constituted a coherent cluster within the family *Geobacteraceae*. Genomic analyses, including average nucleotide identity and DNA–DNA hybridization, clearly differentiated the strains Red96^T^, Red100^T^, and Red88^T^ from other species in the family *Geobacteraceae*, with values below the thresholds for species delineation. Along with the genomic comparison, the chemotaxonomic features further helped distinguish the three isolates from each other. In addition, the lower values of average amino acid identity and percentage of conserved protein, as well as biochemical differences with their relatives, indicated that the three strains represented a novel genus in the family *Geobacteraceae*. Hence, we concluded that strains Red96^T^, Red100^T^, and Red88^T^ represented three novel species of a novel genus in the family *Geobacteraceae*, for which the names *Oryzomonas japonicum* gen. nov., sp. nov., *Oryzomonas sagensis* sp. nov., and *Oryzomonas ruber* sp. nov. are proposed, with type strains Red96^T^ (= NBRC 114286^T^ = MCCC 1K04376^T^), Red100^T^ (= NBRC 114287^T^ = MCCC 1K04377^T^), and Red88^T^ (= MCCC 1K03694^T^ = JCM 33033^T^), respectively.

## 1. Introduction

The family *Geobacteraceae*, belonging to the order *Desulfuromonadales*, is described as consisting of the genera *Geobacter*, *Geomonas*, and a single species *Pelobacter propionicus* [1,2]. Among them, the type genus *Geobacter*, which was first described by Lovley et al. [3] with *Geobacter metalireducens* as the type species, is the most major group, including 17 validated species at the time of writing (Available online: http://www.bacterio.net/geobacter.html). These species are mesophilic and obligate anaerobes, which were usually isolated from terrestrial environments such as forest soil, lotus field mud, freshwater sediments, and oil/metal-contaminated soils [4,5,6]. In addition to the culture-based explorations, the DNA- and RNA-based culture-independent analysis also supports the universal distribution of *Geobacter* in the soils and freshwater sediments [7,8,9]. A common feature among the members of genus *Geobacter* is the ability to reduce Fe(III) to Fe(II) along with the complete oxidation of acetate to CO_2_. Additionally, they can utilize various metals, humic substances, alcohols, hydrogen, organic acids, and aromatic compounds as electron acceptors and/or electron donors, leading to mobilization of metals and mineralization of organic compounds [4]. The ubiquitous distribution and metabolic properties of *Geobacter* species suggest their importance in the biogeochemical cycle of both inorganic and organic materials in the terrestrial and freshwater ecosystems.

Especially in paddy soils, the distribution and predominance of *Geoobacteraceae*, especially *Geobacter*, are reported globally [10,11,12,13,14]. At the beginning of cultivation, paddy fields are waterlogged. Immediately after that, soil redox potential decreases, and an anoxic environment develops within the plowed layer of soil [11]. Such condition induces various biogeochemical processes such as metal reductions, reductive nitrogen transformations, and methanogenesis [11]. Previous studies based on stable-isotope probing and high throughput sequencing, have suggested that *Geobacteraceae*, which dominates in paddy soils, is one of the major drivers directly involved in Fe(III) reduction along with acetate oxidation, nitrogen fixation, dissimilatory nitrate reduction to ammonium (DNRA), and anaerobic ammonium oxidation coupled to Fe(III) reduction (Feammox) and indirectly involved in methanogenesis [8,13,15,16,17,18]. However, limited information is available about the isolation of *Geobacteraceae* strains from paddy soils despite its predominance and pivotal ecological functions [16,19,20], and no paddy soil-derived validated species have been reported except for the genus *Geomonas*, which was recently proposed in our previous study [2]. Although freshwater sediments have prolonged waterlogged conditions unlike paddy soils, our previous study suggests that *Geobacter* is one of the major microbial drivers of the nitrogen cycle in freshwater sediments, as well as paddy soils [13]. However, there are fewer *Geobacter* strains isolated from freshwater sediments compared with microbes involved in the nitrogen cycle, such as nitrogen fixers in other genera [21]. Therefore, we focused on *Geobacteraceae* dominating in paddy soils and freshwater sediments, and try to expand the cultured strains for future studies based on genomics and culturomics [22].

During the screening from paddy soils and the related environments, via the soil slurry incubation as described below, three *Geobacteraceae* strains named Red96^T^, Red100^T^, and Red88^T^ were isolated from two paddy fields and pond sediment, respectively. Comparative 16S rRNA gene sequencing demonstrated that these three novel strains may represent a new taxon of the family *Geobacteraceae*. Therefore, the present study was carried out to define the taxonomy of these three strains through polyphasic characterization.

## 2. Materials and Methods

### 2.1. Isolation and Culture Conditions

The strains Red96^T^ and Red100^T^ were isolated from paddy soils in Nagaoka, Niigata, Japan, and Kanzaki, Saga, Japan, respectively (Appendix A). The strain Red88^T^ was isolated from the pond sediment in Nagaoka, Niigata, Japan (Appendix A). For the isolation of these strains, we applied the following method based upon soil slurry incubation. The air-dried paddy soil, which was collected from the paddy fields in Nagaoka, Niigata, Japan, was suspended in distilled water (soil/water, 1/1.5, *w/v*), and 6 mL of the resulting soil suspension was transferred to a 15 mL serum bottle. The serum bottle was then autoclaved, and 60 µl of Wolfe’s vitamin solution (in l^−1^, 10 mg pyridoxine-HCl, 5 mg thiamine-HCl, 5 mg aminobenzoic acid, 5 mg lipoic acid, 5 mg Ca-pantothenate, 5 mg nicotinic acid, 2 mg biotin, 2 mg folic acid, and 0.01 mg vitamin B_12_) was added. These bottles were sealed with a butyl rubber stopper with an aluminum crimp, and the gaseous phase was exchanged with N_2_/CO_2_ (4:1, *v/v*). Into this soil slurry bottle, 0.1 g raw paddy soil or pond sediment was added as a microbial source and then kept at 30 °C for 2 weeks. After this incubation, 200 µl of the incubated soil suspension was transferred to the fresh soil slurry bottle and cultured at 30 °C for 2 weeks. The resulting cultures were spread on R2A broth (Nihon Pharmaceutical, Tokyo, Japan) medium supplemented with 1.5% agar and 5 mM fumarate (modified R2A plate) and incubated in anaerobic jars at 30 °C for 10 days. The jars were provided with AnaeroPacks (Mistsubishi Gas Chemical, Tokyo, Japan) along with oxygen indicators. After the process was repeated several times, three special isolates, namely Red88^T^, Red96^T^, and Red100^T^, were discriminated from the others by their red colony color. It was found that these three strains also grew well in modified freshwater medium (MFM) supplemented with 20 mM fumarate as the electron acceptor and 20 mM acetate as the electron donor [2,3]. Cultures were stored at −80 °C in MFM supplemented with 10% (*v/v*) of DMSO. *Geobacter chapellei* DSM 13688^T^, obtained from the German Collection of Microorganisms and Cell Cultures (DSMZ), was used as the reference strain throughout the study.

### 2.2. Morphological and Physiological Analysis

The Gram reaction of the three strains was determined using a commercial Gram-staining kit (Sigma, St Louis, MO, USA) according to the manufacturer’s instructions. Cell morphology was observed by transmission electron microscopy (TEM, model JEM-1400, JEOL, Tokyo, Japan) using the cells that were grown for 5 days on modified R2A plates at 30 °C. The temperature range for growth was investigated on modified R2A plates at different temperatures (13, 16, 20, 25, 30, 33, 37, 40, and 42 °C). The pH range for growth was examined between pH 4.5 to 8.5 in increments of 0.5 pH units using R2A broth with 20 mM fumarate (modified R2A broth) at 30 °C supplemented with different pH buffers (acetic acid/acetate, pH 4.5–5.0; MES (20 mM), pH 5.5–6.5; HEPES (20 mM), pH 7.0–8.0; tricine (20 mM), pH 8.5–9.5). For pH stability, all the vials were gassed with helium (He) to reduce the pH fluctuation caused by CO_2_. Growth condition was qualitied with OD_600_ using a spectrophotometer (Jasco V550, Tokyo, Japan) after 3 days of incubation. The effect of NaCl concentration on growth was investigated on modified R2A plates in the presence of 0% to 1.0% (*w/v*) NaCl in increments of 0.1%. Electron acceptor and donor utilization tests were performed using oxygen-free MFM with acetate (10 mM) as the electron donor for all electron acceptor tests and Fe(III)-NTA (10 mM) as the electron acceptor for all electron donor tests. The concentrations of tested electron acceptors and donors were utilized as described by Nevin et al. [23]. Cytochrome *c* analysis was carried out using the whole cells grown in degassed MFM with 20 mM acetate and 10 mM fumarate. The dithionite-reduced minus air-oxidized difference spectrum was generated on a Jasco V550 spectrophotometer [3]. Various enzyme activities of the three isolates and the reference strain were determined using API ZYM strips (bioMérieux, Lyon, France) in accordance with the manufacturer’s instructions. Fe(III)-NTA (5 mM) and Fe(III) citrate (20 mM) were used as the substrate to determine the iron reduction curve. The concentration of ferrous irons at different culture time was measured using the ferrozine assay at 562 nm with a Jasco V550 spectrophotometer, as described previously [24]. 

### 2.3. Phylogenetic Analysis

The nearly complete 16S rRNA genes of the three isolates were amplified and sequenced with the primers 27F/1492R and 27F/926R/1492R, respectively, as previously described by Ashida et al. [25]. The alignments and calculations of similarity values of the almost complete 16S rRNA gene sequences corresponding to three strains were carried out using EZBioCloud’s Identify service [26]. The phylogenetic trees were constructed using the MEGA 7.0 by implementing the neighbor-joining (NJ), maximum-likelihood (ML), and maximum-parsimony (MP) methods with 1000 bootstrap replicates [27]. Multilocus sequence analysis (MLSA) [28] of the partial sequences of five housekeeping genes (*rpoB, gyrB, recA, fusA,* and *nifD*) was performed as described by Holmes et al. [29] with the primers NIFDF2/NIFDR2, GYR48F/GYR1010R, RPO175F/RPO800R, REC300F/REC910R, and a modified primer pair for *fusA* (Forward: 5′-CTNGACATCAAGATCTGCCC-3′, Reverse: 5′-TTCGCCTCNACCTTGAACTC-3′). Subsequently, the nucleic acid sequences were translated to amino acid sequences according to the standard codons. Phylogenetic analysis of these deduced amino acid sequences was performed using the MEGA 7.0 [27] by implementing the ML method based upon the best-fit substitution model after alignment and concatenation of these five housekeeping genes.

### 2.4. Genome Sequencing, Assembly and Annotation

Genomic DNA was extracted and purified using DNeasy blood and tissue kit (Qiagen, Hilden, Germany) in accordance with the manufacturer’s instructions. The draft genome sequencing of the three strains was performed using the Illumina HiSeq instrument according to the manufacturer’s instructions (Illumina, San Diego, CA, USA) for 2 × 150 paired-end (PE) configuration at Genewiz Inc. (Suzhou, China). The resulting sequence reads were quality tested and then assembled using the Velvet (version: 1.2.10, available online: https://www.ebi.ac.uk/~zerbino/velvet/) [30]. The gaps were filled using SSPACE [31] and GapFiller [32]. Assembled contigs were used for G + C content calculation and preliminary gene annotation by RAST [33] and NCBI. Translated amino acids were further assigned to KEGG pathways with the BlastKOALA server [34] and the eggNOG database using eggNOG-mapper 2 [35]. All of the reference genomes used in this study were obtained from the NCBI database and analyzed equally with the three isolated strains. All the sequences were deposited in the GenBank database and the numbers are available in the Appendix B.

### 2.5. Genome Comparison

The average nucleotide identity (ANI) and the digital DNA-DNA hybridization (dDDH) between the three strains and other type species in the family *Geobacteraceae* were obtained in silico using the JspeciesWS based upon the BLAST+ [36] and the Genome-to-Genome Distance Calculator 2.1 (GGDC) with recommended BLAST+ alignment and Formula 2 [37]. The average amino acid identity (AAI) and percentage of conserved protein (POCP) were determined based upon annotated protein sequences as described previously [38,39] using an online-tool (Available online: http://enve-omics.ce.gatech.edu/) and a Python script (Available online: https://github.com/2015qyliang/POCP), respectively. Phylogenetic tree based on the whole genome sequences was constructed using an up-to-date bacterial core gene (UBCG) pipeline (Available online: https://www.ezbiocloud.net/tools/ubcg) with the default parameters as described by Na et al. [40]. A visual genomic comparison at the nucleotide level of the three isolated strains and their relatives was carried out by the BLAST ring image generator (BRIG) with default parameters [41]. To display the locally collinear blocks (LCBs) and their similarities, the final alignments of the three strains were performed by Mauve [42] using default parameters, and then a diagrammatic representation of these alignments was generated using genoPlotR [43]. Analysis of the core genome of the three isolated was performed using the Roary (version: 3.6.0, available online: https://github.com/sanger-pathogens/Roary) [44] along with the rapid genome annotation pipeline Prokka (version: 1.11, available online: https://github.com/tseemann/prokka) [45]. Both of them were run with default parameters.

### 2.6. Genomic Fingerprints

For the specific fingerprints, total genomic DNA of the three isolates was used as a template for PCR amplification by the methods of random amplified polymorphic DNA (RAPD) (primer: AGCAGCGTGG) [46] and repetitive sequence-based PCR (rep-PCR) with ERIC primers (ERIC1/ERIC2) and BOX primer (BoxA1R) [47]. Then 20 uL of the PCR products were electrophoresed in a 1.5% agarose gel using 1 × TAE buffer mixed with 3.0 μL of ethidium bromide/mL under 100 V for 45 min.

### 2.7. Chemotaxonomic Characterization

Fatty acid detection was performed using the cells grown in modified R2A broth which were incubated at 30 °C. The strains Red96^T^, Red100^T,^ and Red88^T^ were cultured for 2 days, while the reference strain *G. chapellei* DSM 13688^T^ was cultured for 4 days due to the different culture time to their late exponential growth phases. Fatty acids were extracted and methylated in accordance with the protocol described by Kuykendall et al. [48]. Subsequently, the fatty acid compositions were monitored using a gas chromatography mass spectrum (GCMS-QP2010 ultra, Shimadzu, Kyoto, Japan) with the different known fatty acids as the standard [49]. Quinones were extracted from fresh biomass grown in modified R2A broth for 5 days and purified as described previously [50]. Quinone profiles were determined by high-performance liquid chromatography (HPLC) with ACQUITY UPLC H-Class system (Waters, Milford, MA, USA) by TechnoSuruga Laboratory (Shizuoka, Japan). 

## 3. Results and Discussion

### 3.1. Morphological and Physiological Analysis

The three identified strains, namely Red96^T^, Red100^T^, and Red88^T^, were Gram-negative, strictly anaerobic, rod-shaped, and slightly curved in appearance (Figure 1). Cells were 0.4–0.8 µm wide, 1.0–3.0 µm long, and motile due to the presence of peritrichous flagella (Figure 1). The colonies grown on modified R2A plates after 5 days of incubation were red-pigmented, circular, smooth, and less than 1.5 mm in diameter (Figure 1). The three strains also showed a common growth condition such temperature ranging from 16–40 °C (optimum 30–33 °C), pH ranging from 5.0–7.5, and an optimal NaCl concentration of 0%–0.1%. Additionally, the optimal pH for strains Red96^T^ and Red88^T^ was 5.5–6.5, and 6.0–7.0 for strain Red100^T^ indicating the three strains to be slightly acidophilic. Nitrate reduction occurred for strain Red88^T^ with ammonium production, however, no such activity was observed for the strains Red96^T^ and Red100^T^. The dithionite-reduced minus air-oxidized difference spectrum of all analyzed strains had peaks of *c*-type cytochromes. The absorption peaks for strain R88^T^ were 418, 522, and 552 nm, strains Red96^T^ and Red100^T^ were 424, 524, and 554 nm, while the reference strain *G.*
*chapellei* DSM 13688^T^ were 423, 526, and 553 nm (Appendix A). The spectra obtained for Red88^T^ were the same as that of *G. metallireducens* GS-15^T^ with absorption peaks at 418 nm, 522 nm, and 552 nm [51] and were entirely different from those obtained for Red96^T^ and Red100^T^. Based on API ZYM strips, the three isolates shared same enzymatic activities, they are positive for alkaline phosphatase, leucine arylamidase, acid phosphatase, and naphthol-AS-BI-phosphohydrolase activities, while the reference strain *G.*
*chapellei* DSM 13688^T^ was positive for esterase (C4), acid phosphatase, esterase lipase (C8), and naphthol-AS-BI-phosphohydrolase activities. The three isolates had ferric-reducing ability similar to other species belonging to the family *Geobacteraceae*. Although the ferric reducing rate of three isolates with two different ferric types were broad, most ferric irons were reduced to ferrous irons in 10 days (Figure 2). More detailed phenotypic features of these three novel strains were described in Table 1 and the species description.

### 3.2. Phylogenetic Analysis

Phylogenetic analysis based upon the nearly complete 16S rRNA gene sequences of strains Red96^T^ (1428 bp, accession number MK334373), Red100^T^ (1429 bp, accession number MK334374), and Red88^T^ (1429 bp, accession number MK334372) demonstrated that the three strains belonged to the family *Geobacteraceae* in the class *Deltaproteobacteria* and formed a coherent branch near to *Geobacter lovleyi* SZ^T^ and *Geobacter thiogenes* K1^T^ (Figure 3, Appendix A). On the basis of 16S rRNA gene sequence similarity, the three strains were closely related to *P. propionicus* Ott Bd 1^T^, *G. lovleyi* SZ^T^, *G. chapellei* 172^T^, and *G. thiogenes* K1^T^ with similarity values ranging from 95.36% to 95.73% (Appendix A). The pairwise sequence similarity was observed to be in the range of 99.37% to 99.79% for the three strains (Appendix A). Phylogenetic analysis of MLSA also showed that the three isolates formed an independent group within the family *Geobacteraceae* (Figure 4), indicating a robust phylogenetic relationship of the three strains. 

Although the three strains were considered as different taxa in the family *Geobacteraceae,* they showed low 16S rRNA gene similarities to the two type species, *G. metallireducens* GS-15^T^ and *Geomonas oryzae* S43^T^, with values in the range of 92.22%–92.43%, and 93.54%–93.75%, respectively (Appendix A). Moreover, Yarza et al. [56] have proposed that the generally used arbitrary genus threshold of 95% 16S rRNA gene identity should be revised to 94.5% with a confidence interval of 94.55–95.05 and median sequence identity of 96.4%. Along with the monophyletic cluster in phylogenetic trees, we hence assessed that the three strains are not the members of two proposed genera and may represent a novel genus in the family *Geobacteraceae*.

### 3.3. Genome Characteristics

The sizes of the draft genomes of the three strains Red96^T^, Red100^T^, and Red88^T^ were 3.6, 3.6, and 3.8 Mb contained 30, 18, and 23 contigs (>500 bp) with N50 of 336264, 620468, and 353860 bp, respectively. For strains Red96^T^, Red100^T^, and Red88^T^, a total of 3363, 3322, and 3523 potential coding sequences were predicted by RAST, respectively. Among them, 2224, 2092, and 2208 protein-coding genes were annotated and classified into different functional groups based on the SEED database (Appendix A). Besides, 3000, 2942, and 3060 genes with COG numbers were annotated using the eggNOG database (Appendix A), while 1426, 1412, and 1392 genes with ko numbers were annotated with the KEGG database (Appendix A). More detailed genomic features were listed in Table 2. The average G + C content of the strains Red96^T^, Red 100^T^, and R88^T^ were 59.0, 59.7, and 58.4 mol%, respectively, markedly higher than those of the reference strain, *G. chapellei* DSM 13688^T^ (50.2 mol%) [55], as well as other relatives *G. lovleyi* SZ^T^ (54.8 mol% from genome) and *G. thiogenes* K1^T^ (52.8 mol% from genome), but comparable to their another close neighbour *P. propionicus* Ott Bd 1^T^ (58.5 mol% from genome). 

As described above, strains Red96^T^, Red100^T^, and Red88^T^ exhibited the activity of Fe(III) oxide reduction (Figure 2). There are two known bacterial gene clusters involved in the ferric reduction; the porin-cytochrome genes (*pcc*) cluster and metal-reducing gene (*mtr*) cluster, which have originally elucidated in *Geobacter sulfurreducens* and *Shewanella oneidensis*, respectively [57]. In genomes of strains Red96^T^, Red100^T^, and Red88^T^, we found *pcc* homologous including constitutive genes encoding outer membrane- (OmcB/C–OmbB/C–OmaB/C), periplasmic- (PpcA), and cytoplasmic membrane- proteins (CbcL) (Appendix A), suggesting that these three strains reduce Fe(III) oxide by using the porin-cytochrome system as well as *Geobacter* spp. [58,59]. Moreover, their genomes also harbored genes involved in the synthesis of electrically conductive pili (e-pili), including *pilA* coding an e-pili assembler (Appendix A, [60]). e-pili plays a key role in long-range extracellular electron transport, which allows for an electron exchange with farther minerals and microorganisms [61]. Therefore, from the viewpoint of efficient electron transfer, microorganisms having e-pili are popular research targets in the field of bioelectrochemistry and biogeochemistry, such as the development of microbial fuel cells and the bioremediation of heavy metal contaminations [62]. However, e-pili synthesis genes have been found in the genomes of limited bacterial groups [63]. Strains Red96^T^, Red100^T^, and Red88^T^ would be novel candidates as valuable microbial resources in applied biochemistry research.

### 3.4. Genome Comparison

To carry out a complete phylogenetic analysis, a phylogenetic tree based on 92 concatenated core genes (UBCG) was also constructed. This phylogenetic tree confirmed strains Red96^T^, Red100^T^, and Red88^T^ formed a novel robust cluster in the family *Geobacteraceae* (Figure 5), supporting their distinct positions with other known species. However, the phylogenetic clades in UBCG trees differed slightly in comparison to the trees which were constructed based upon the 16S rRNA sequences and MLSA (Figure 3 and Figure 4, Appendix A). In the UBCG tree, the strain Red96^T^ had a closer relationship with strain Red88^T^ (Figure 5); however, it was observed to be closely related to the strain Red100^T^ based upon the 16S rRNA gene and MLSA analysis (Figure 3 and Figure 4, Appendix A). Moreover, the discrepancy also occurred between the phylogenetically coherent branches of the three strains and their relatives for three differently constructed trees. The three isolated strains were closely related to *G. lovleyi* and *G. thiogenes* in the 16S rRNA gene and MLSA trees (Figure 3 and Figure 4, Appendix A) as compared to that of the UBCG tree where *P. propionicus* was observed to the closest relative (Figure 5). These conflicting results indicate that strains Red96^T^, Red100^T^, and Red88^T^ may represent a separated taxon, including three independent subtaxa in the family *Geobacteraceae*.

The ANI and GGDC values were below 74.0% and 22.2%, respectively, between the strains R88^T^, Red96^T^, and Red 100^T^ and other *Geobacteraceae* species (Table 3). According to the recommended species cut-off values for the ANI and GGDC (95%–96% and 70%, respectively) [64], the three isolates were suggested to be a novel taxon in the family *Geobacteraceae*. Furthermore, the ANI and GGDC values ranged from 94.2%–95.6% and 58.2%–67.5%, respectively, between each pair of the three isolates (Table 3). Among them, strain Red100^T^ could be clearly separated as a novel species from Red96^T^ and Red88^T^, owing to the lower ANI and GGDC values than the recommended threshold, as described above. However, slightly more variability was observed for the strains Red88^T^ and Red96^T^ because the genomic relatedness between them was 95.6% for ANI and 67.5% for GGDC (Table. 3), which were in the transition zone for proposing novel species: 95%–96% for ANI and 60%–70% for GGDC [65]. 

The maximum values of AAI and POCP were observed to be 66.6% and 61.6%, respectively, between the three isolated strains and other *Geobacteraceae* species (Table 3). These values were below the threshold to separate recently-proposed genus *Geomonas* from other *Geobacteraceae* species: ca. 70% for AAI and ca. 65% for POCP (Figure 6 [2]). Previous reports indicated that the AAI and POCP, based on the amino acid sequences comparison, are robust approaches to determine the bacterial genera separation [39,66]. Together with the above 16S rRNA gene similarities and phylogenetic analysis, we concluded that the three analyzed strains represent a novel genus in the family *Geobacteraceae*. 

Additionally, the BRIG analysis suggested that most genomic regions of the five analyzed strains were conserved with at least 70% identity (Figure 7). Moreover, strains Red96^T^, Red100^T^, and Red88^T^ showed higher identity with each other compared with the two reference strains in most genomic regions, implying that the three isolated strains are much closer to each other than to *G. lovleyi* and *P. propionicus*. Based on the core genomes, most genes (ca. 60%) were found shared by the three strains, and the less (ca. 20%) were unique genes (Figure 8), showing the comparability and individuality among them. Detailed analysis also revealed that strain Red96^T^ shared more genes with strain Red88^T^ (412), but less with strain Red100^T^ (93), indicating strain Red96^T^ had a closer relationship with strain Red88^T^, which is consistent with genomic phylogenetic analysis, but dissimilar with the guide tree of Mauve alignment (Figure 9), which showed strains Red96^T^ and Red100^T^ are more closely related to each other than to strain Red88^T^. Besides that, the Mauve alignment also displayed the presence of all LCBs with partial regions of inversion and rearrangement, as well as a strong synteny among these three strains (Figure 9). Although the guide tree of Mauve alignment is distinct from genomic analysis (Figure 5 and Figure 9; Table 3), it agrees with the phylogenetic analysis of MLSA and 16S rRNA (Figure 3 and Figure 4, Appendix A). This conflicting result further proved the individual status of these three strains.

### 3.5. Genomic Fingerprints

The banding patterns obtained using rep-PCR to fingerprint the three isolates have are shown in Appendix A. The banding patterns of the three isolates were obviously different from each other, indicating that the three isolates were not of clonal origin and were clearly different from each other.

### 3.6. Chemotaxonomic Characterization

The fatty acid profiles of the three isolates and the reference strain *G.*
*chapellei* DSM 13688^T^ are shown in Table 4. Predominant fatty acids of the three isolates were identified as iso-C_15:0_ and C_16:1_
*ω7c*, with an aggregate proportion accounting for more than 50% of all fatty acids. Besides, the strains Red100^T^ and Red88^T^ also contained C_16:0_ in greater proportions, which were 10.8% and 13.8%, respectively. C_16:1_
*ω7c*, C_16:0_, and iso-C_15:0_ were determined as the predominant fatty acids (relative account > 10%) for the reference strain *G.*
*chapellei* DSM 13688^T^ showing similar compositions to those of the three strains, however, the relative proportions distinguished them clearly. Other minor fatty acid profiles of these strains show no significant differences. The predominant respiratory quinone of the three strains was MK-8, in agreement with the major quinone component of other *Geobacteraceae* species [2,5,6]. 

## 4. Conclusions

In this study, three *Geobacter*-related strains, namely Red96^T^, Red100^T^, and Red88^T^, were isolated from the paddy soils and pond sediment in Japan. Taxonomic and phylogenetic analysis of the nearly complete 16S rRNA gene sequences and five concatenated housekeeping genes revealed that the three isolated strains were most similar to the species in the family *Geobacteraceae*, however, they shared lower 16S rRNA similarities with the two type species, *G. metallireducens* GS-15^T^ and *G. oryzae* S43^T^. Moreover, the three strains showed a remarkable physiological difference from their close neighbour *P. propionicus* Ott Bd 1^T^, because *P. propionicus* can neither utilize acetate as an electron donor nor oxidizes organic compounds completely [52]. Additionally, unlike the three isolated strains, *P. propionicus* did not contain the *c*-type cytochromes that are involved in electron transfer to Fe(III) [52]. Meanwhile, these three strains also differed from the other two neighbors, *Geobacter lovleyi* SZ^T^ and *Geobacter thiogenes* K1^T^, because both of them were tested in this study that they cannot grow using R2A agar plates or broth with 5mM fumarate. Genomic analysis based on the ANI and GGDC values indicated the three strains represented at least two novel species (Red96^T^ and Red100^T^) with the values below the threshold for species separation, though the status of Red88^T^ could not be determined. Furthermore, the AAI and POCP showed lower values between the three strains and their relatives than the delineation proposed for genera separation in the family *Geobacteraceae*. These facts suggest that the three strains represent a novel genus in the family *Geobacteraceae*.

Although the strain Red88^T^ could not be separated clearly from Red96^T^, owing to the high genomic relatedness located in the transition zone for novel species description, their biochemical and physiological characteristics, including the optimal growth conditions, absorption peaks of *c*-type cytochromes, alternative electron donors/acceptors, and fatty acid profiles, clearly differed. Furthermore, the inconsistent phylogenetic positions of the three strains in the three phylogenetic trees also suggests the independent statuses of the three isolated strains, as also supported by the core genome and rep-PCR analysis. Thus, based upon the phenotypic, chemotaxonomic, phylogenetic, and genomic characterizations performed in this study, we concluded that these three strains represented three novel species of a novel genus in the family *Geobacteraceae*. Hence, we proposed a novel genus, *Oryzomonas* gen. nov. sp. nov., comprising three novel species: *Oryzomonas japonicum* sp. nov., *Oryzomonas sagensis* sp. nov., and *Oryzomonas ruber* sp. nov.

Furthermore, functional tests revealed that these three strains have the ferric reducing ability, and the genomic analysis suggested that they have the long-range electron transportable capacity using their e-pili. As reported previously [4], such capacity enables the efficiency of electron transportation between metal compounds to improve, leading to metal-reducing microorganisms with e-pili being of a great significance in bioremediation of hazardous metals such as U, Se, As, and Cr, and development of microbial fuel cells [4]. Hence, this study not only improves our understanding for the diversity of important environmental microorganisms but also provides novel valuable microbial resources in applied biogeochemical science.

Together with our previous study [2], 6 type species (4 *Geomonas* and 2 *Oryzomonas*) of the family *Geobacteraceae* were isolated from paddy soil and revealed to be phylogenetically derivative but distinct from traditional *Geobacter* (Figure 3, Figure 4 and Figure 5). Although the substantial research based on culture-independent methods indicate the predominance of *Geobacter* in paddy soils as described above, *Geomonas* and *Oryzomonas* have been probably confined to ‘*Geobacter*’ in such previous research. In order to improve our understanding of diversity of predominant members in paddy soils, a careful phylogenetic analysis should be done, taking into account the *Geomonas* and *Oryzomonas*, in data sets obtained from previous and future culture-independent approaches. 

**Description of *Oryzomonas* gen. nov.:***Oryzomonas* (O.ry.zo.mo’nas. Gr. fem. n. *oryza*, rice; L. fem. n. *monas*, a unit, monad; N.L. fem. n. *Oryzomonas*, a monad from rice soil). Cells are strictly anaerobic, Gram-negative, and rod-shaped. Colonies on R2A agar plates supplementary with 5 mM fumarate are smooth, spherical and red-pigmented. Able to use Fe(III) citrate, Fe(III)-NTA, ferrihydrite and fumarate as the electron acceptors along with a variety of different electron donors, including yeast powder, acetate, tryptone, succinate, glucose, pyruvate, glycerol, propionate, methanol, and malate. The predominant fatty acids are iso-C_15:0_, and C_16:1_
*ω7c*. The major quinone is MK-8. The type species is *Oryzomonas japonicum*.

**Description of *Oryzomonas japonicum* sp. nov.:***Oryzomonas japonicum* (ja.po’ni.ca. N.L. fem. adj. *japonica* Japanese, pertaining to Japan). Displays the following properties besides those given in the genus description. Cells are rod shaped with light curve in appearance (0.4–0.8 µm wide, 1.0–3.0 µm long) and motile with peritrichous flagella. Strains display good growth on R2A agar plus 5–20 mM fumarate and freshwater medium. Growth occurs at 16–40 °C (optimum, 30–33 °C), at pH 5.0–7.5 (optimum, 5.5–6.5), and with 0%–0.7% (*w/v*) NaCl (optimum, 0–0.1%). Can also utilize lactate, mannitol, ethanol, casamino acid, nicotinate, glutamine, arginine, serine, and proline as electron donors in the presence of Fe(III)-NTA as electron acceptor, but not phenol, isopropanol, toluene, butanol, benzaldehyde or benzyl alcohol. Can also utilize Fe(III) pyrophosphate, Fe(III)-EDTA and MnO_2_ as electron acceptors in the presence of acetate as electron donor, but not sulfur, nitrite or nitrate. Alkaline phosphatase, acid phosphatase, leucine arylamidase, and naphthol-AS-BI-phosphohydrolase activities were present but valine arylamidase, esterase (C4), trypsin, esteraselipase (C8), lipase (C14), cystine arylamidase, α-glucosidase, α-chymotrypsin, α-galactosidase, α-mannosidase, β-galactosidase, β-glucuronidase, β-glucosidase, N-acetyl-β-glucosaminidase, and α-fucosidase activities are absent. The predominant fatty acids are iso-C_15:0_ and C_16:1_
*ω7c*. The type strain, Red96^T^ (=NBRC 114286^T^ = MCCC 1K04376^T^), was isolated from paddy soil collected from Niigata, Japan. The DNA G+C content of type strain is 59.0 mol%.

**Description of *Oryzomonas sagensis* sp. nov.:***Oryzomonas sagensis* (sa.gen’sis. N.L. fem. adj. *sagensis,* of or pertaining to the Saga Prefecture of Japan). Displays the following properties besides those given in the genus description. Cells are rod shaped with light curve in appearance (0.4–0.8 µm wide, 1.0–2.5 µm long) and motile with peritrichous flagella. Strains display good growth on R2A agar plus 5–20 mM fumarate and freshwater medium. Growth occurs at 16–40 °C (optimum, 30–33 °C), at pH 5.0–7.5 (optimum, 6.0–7.0), and with 0%–0.5% (*w/v*) NaCl (optimum, 0%–0.1%). Can also utilize glutamine, mannitol, nicotinate, casamino acid, arginine, serine, and proline as electron donors in the presence of Fe(III)-NTA as the electron acceptor, but not toluene, isopropanol, phenol, lactate, butanol, benzyl alcohol, ethanol, or benzaldehyde. Can also utilize Fe(III)-EDTA and Fe(III) pyrophosphate as electron acceptor in the presence of acetate as electron donor, but not sulfur, MnO_2_, nitrite or nitrate. Alkaline phosphatase, naphthol-AS-BI-phosphohydrolase, acid phosphatase and leucine arylamidase activities were present but esterase (C4), trypsin, lipase (C14), valine arylamidase, esteraselipase (C8), cystine arylamidase, α-chymotrypsin, α-glucosidase, α-fucosidase, α-galactosidase, α-mannosidase, β-galactosidase, β-glucosidase, β-glucuronidase, and N-acetyl-β-glucosaminidase activities are absent. The predominant fatty acids are iso-C_15:0_, C_16:1_*ω7c* and C_16:0_. The type strain, Red100^T^ (= NBRC 114287^T^ = MCCC 1K04377^T^), was isolated from paddy soil collected from Saga, Japan. The DNA G+C content of type strain is 59.7 mol%.

**Description of *Oryzomonas ruber* sp. nov.:***Oryzomonas ruber* (ru’bra. L. fem. adj. *rubra*, red). Displays the following properties besides those given in the genus description. Cells are rod shaped with light curve in appearance (0.4–0.8 µm wide, 1.0–3.0 µm long) and motile with peritrichous flagella. Strains display good growth on R2A agar supplemented with 5–20 mM fumarate and freshwater medium. Growth occurs at 16–40 °C (optimum, 30–33 °C), at pH 5.0–7.5 (optimum, 5.5–6.5), and with 0%–0.5% (*w/v*) NaCl (optimum, 0%–0.1%). Can also utilize mannitol, lactate, nicotinate, casamino acid, ethanol, glutamine, arginine, serine, and proline as electron donors in the presence of Fe(III)-NTA as electron acceptor, but not phenol, isopropanol, benzyl alcohol, butanol, benzaldehyde or toluene. Can also utilize Fe(III)-EDTA, Fe(III) pyrophosphate, MnO_2_ and nitrate as electron acceptor in the presence of acetate as electron donor, but not sulfur. Alkaline phosphatase, naphthol-AS-BI-phosphohydrolase, acid phosphatase and leucine arylamidase activities were present but esterase (C4), trypsin, lipase (C14), valine arylamidase, esteraselipase (C8), cystine arylamidase, α-chymotrypsin, α-fucosidase, α-glucosidase, α-galactosidase, α-mannosidase, β-glucuronidase, β-galactosidase, β-glucosidase, and N-acetyl-β-glucosaminidase activities are absent. The predominant fatty acids are iso-C_15:0_, C_16:1_
*ω7c* and C_16:0_. The type strain, Red88^T^ (= MCCC 1K03694^T^ = JCM 33033^T^), was isolated from sediment of a pond in Niigata, Japan. The DNA G+C content of type strain is 58.4 mol%.

## Figures and Tables

**Figure 1 microorganisms-08-00634-f001:**
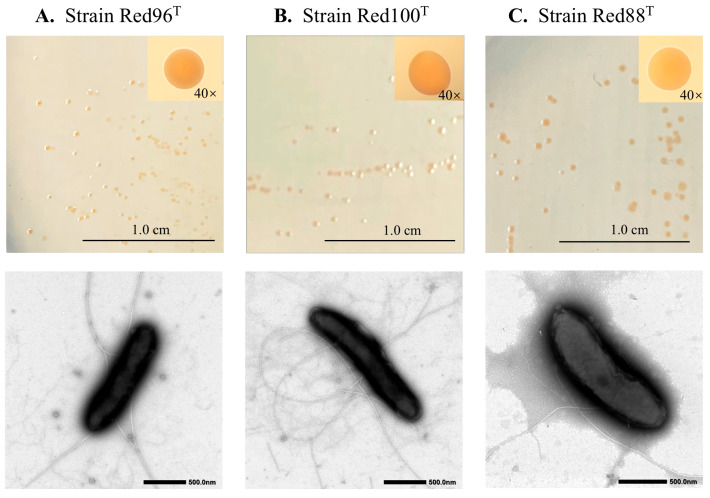
Colony and cell morphology of three isolated strains: *Oryzomonas japonicum* Red96^T^ (**A**), *O. sagensis* Red100^T^ (**B**), and *O. ruber* R88^T^ (**C**). Upper and bottom panels indicate colonies on the modified R2A plate and TEM image of the cells, respectively.

**Figure 2 microorganisms-08-00634-f002:**
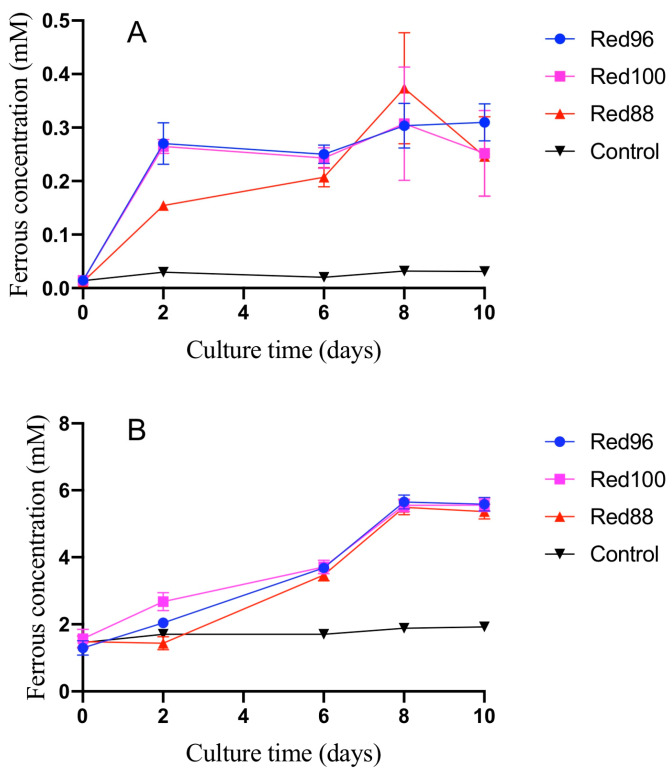
Reduction of insoluble Fe(III) by the three novel strains: *Oryzomonas japonicum* Red96^T^, *O. sagensis* Red100^T^, and *O. ruber* R88^T^. The experiments were performed using MFW medium with 10 mM acetate as the electron donor. (**A**) Soluble ferrous concentration from Fe(III)-NTA reduction; (**B**) soluble ferrous concentration from Fe(III) citrate reduction. Data derived from triplicate tests were presented as means ± standard deviations (SD).

**Figure 3 microorganisms-08-00634-f003:**
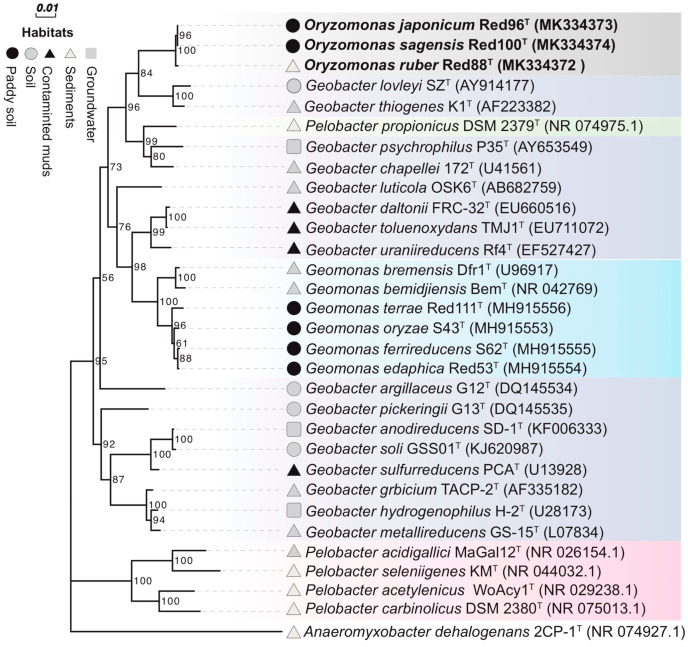
Phylogenetic tree of strains Red96^T^, Red100^T^, Red88^T^, and representatives in the order *Desulfuromonadales* based on 16S rRNA sequence divergence. The tree was inferred by the neighbor-joining (NJ) algorithm using MEGA 7.0 with Kimura 2-parameter model. Habitats, representing the isolation source of the type strains, are coded by different marks. The background colors represent different bacterial genera. Bootstrap values (expressed as percentages of 1000 replications) over 50% are shown at branching nodes. Bar, 0.01 substitutions per nucleotide position.

**Figure 4 microorganisms-08-00634-f004:**
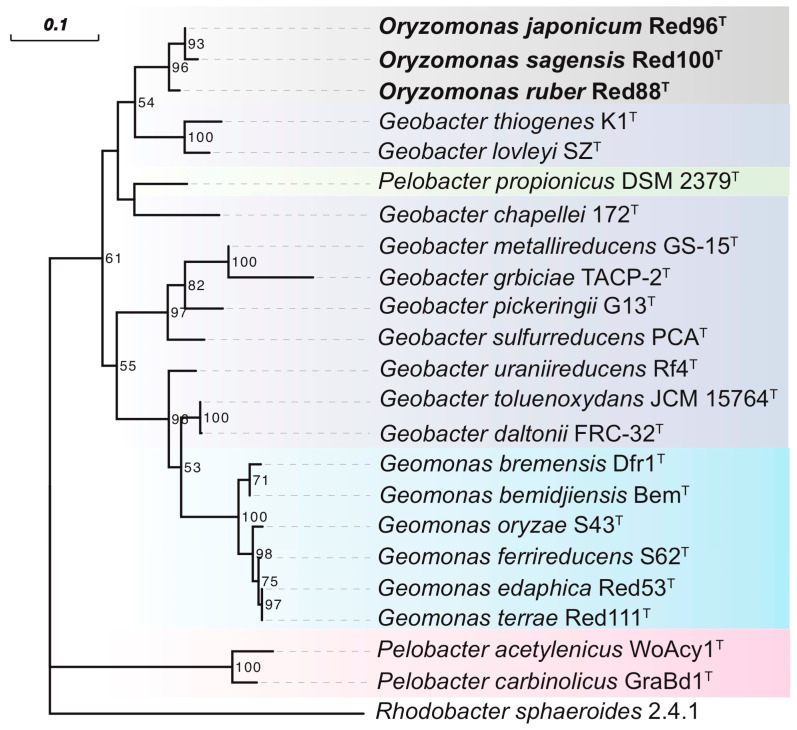
Phylogenetic tree of strains Red96^T^, Red100^T^, Red88^T^, and representatives in the order *Desulfuromonadales* based on deduced protein sequences of five concatenated housekeeping gene: *fusA* (1–196 amino acids), *gyrB* (197–491 amino acids), *nifD* (492–752 amino acids), *recA* (753–977 amino acids) and *rpoB* (978–1177 amino acids). The background colors represent different bacterial genera. The tree was inferred by maximum-likelihood (ML) with the best-fit substitution model (LG + G), and 1000 bootstrap replicates. Over 50% are shown at branching nodes. Bar, 0.1 substitutions per nucleotide position.

**Figure 5 microorganisms-08-00634-f005:**
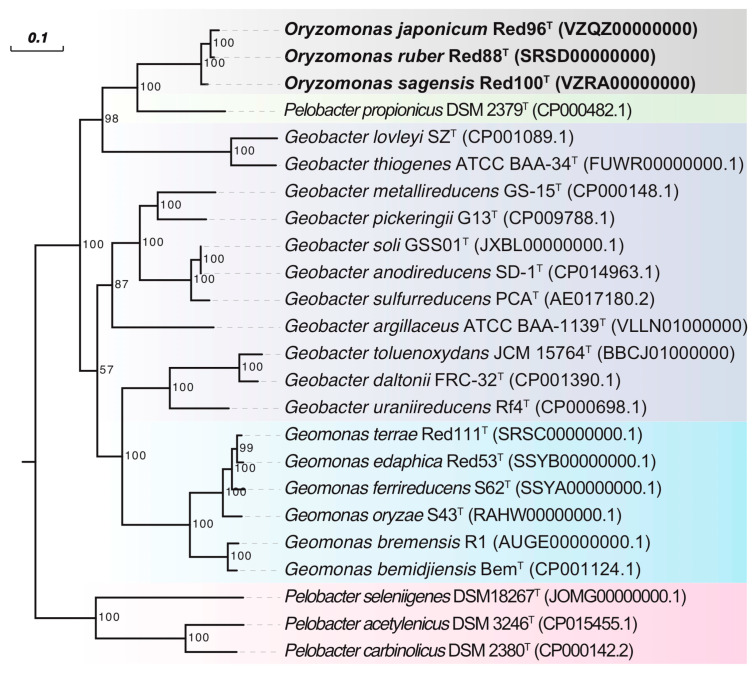
Phylogenetic tree of strains Red96^T^, Red100^T^, Red88^T^, and representatives in the order *Desulfuromonadales* based on the whole genome sequences. The tree was reconstructed based on a concatenated alignment of 92 core genes using RAxML tool with GTR + CAT model [40]. The background colors represent different bacterial genera. Bootstrap values (expressed as percentages of 100 replications) over 50% are shown at branching nodes. Bar, 0.1 substitutions per nucleotide position.

**Figure 6 microorganisms-08-00634-f006:**
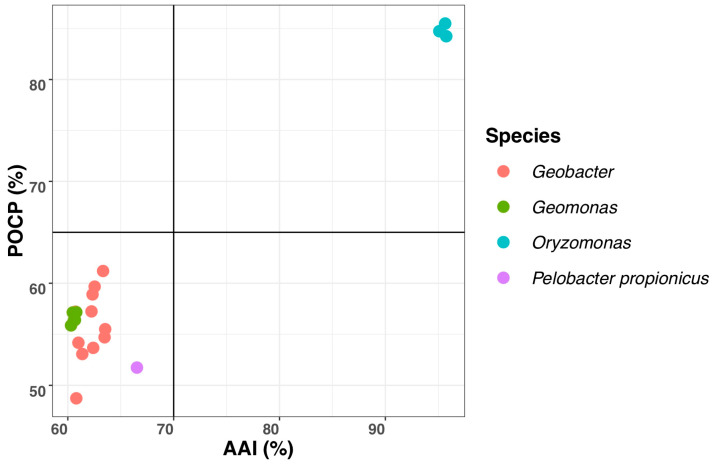
Comparisons of genomic similarities based on AAI and POCP values. The points represent average comparison values between the three novel strains and their close relatives in the family *Geobacteraceae*. A total of 20 genomes were included in this analysis.

**Figure 7 microorganisms-08-00634-f007:**
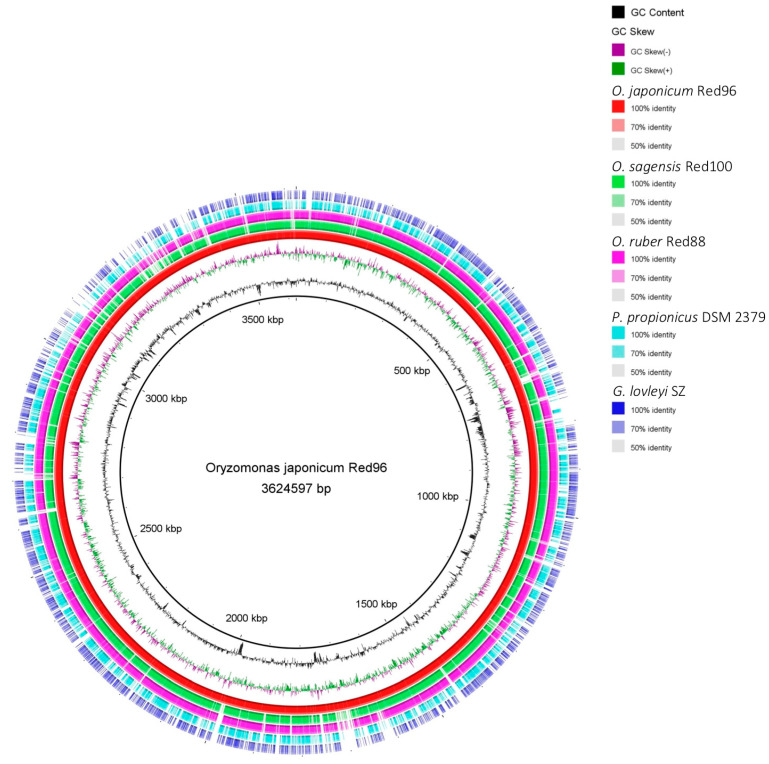
Circular representation of whole-genome sequences of *Oryzomonas japonicum* Red96^T^, *O. sagensis* Red100^T^, *O. ruber* R88^T^, and their two neighbors: *Pelobacter propionicus* DSM 2379^T^ and *Geobacter lovleyi* SZ^T^. The rings from inner to outer: ring 1–GC content, ring 2–GC skew, ring 3–whole-genome sequences of *O. japonicum* Red96^T^ (red), ring 4–whole-genome sequences of *O. sagensis* Red100^T^ (green), ring 5–whole-genome sequences of *O. ruber* R88^T^ (pink), ring 6–whole genome sequences of *P. propionicus* DSM 2379^T^ (cyan), ring 7–whole-genome sequences of *G. lovleyi* SZ^T^ (blue). The genome of *O. japonicum* Red96^T^ was used as the reference for global comparisons.

**Figure 8 microorganisms-08-00634-f008:**
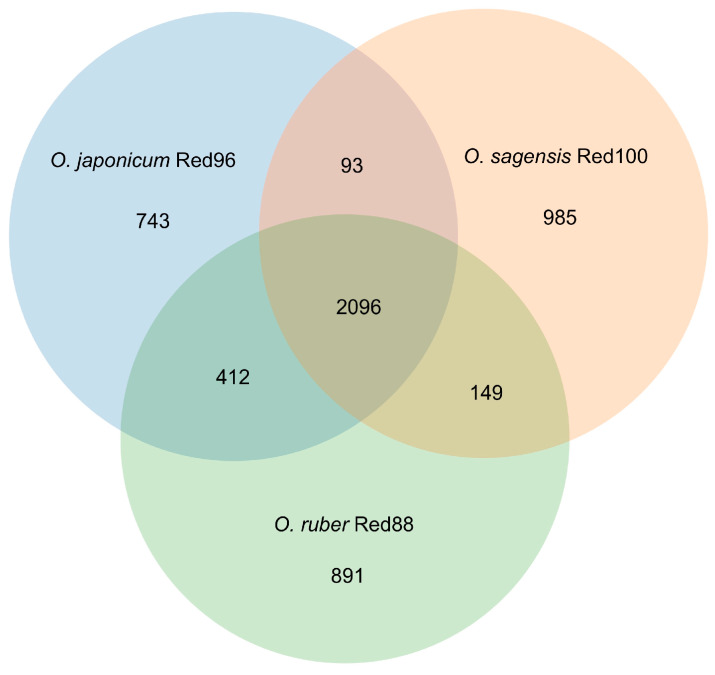
Venn diagram of homologs showing the number of shared coding regions among the three strains *Oryzomonas japonicum* Red96^T^ (blue), *O. sagensis* Red100^T^ (red), and *O. ruber* R88^T^ (green).

**Figure 9 microorganisms-08-00634-f009:**
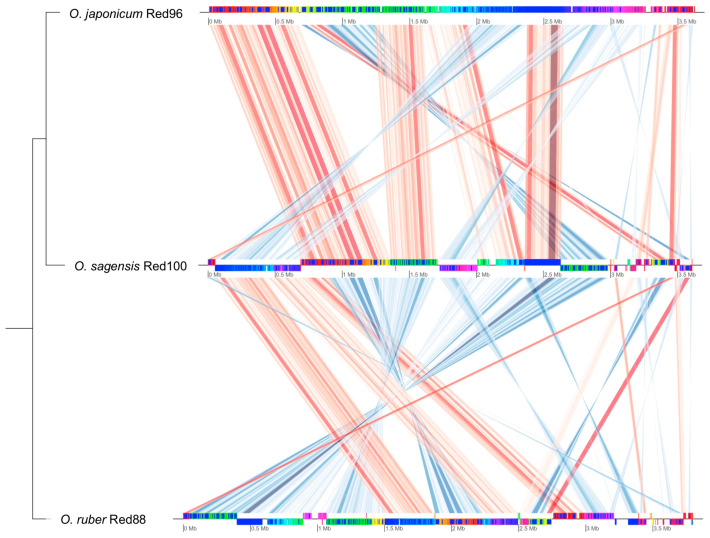
Multiple genome alignment of the three novel strains *Oryzomonas japonicum* Red96^T^ (top), *O. sagensis* Red100^T^ (middle), and *O. ruber* R88^T^ (bottom). The genome of *O. japonicum* Red96^T^ was used as the reference for global alignment. Lines connect the homologous local conserved blocks (LCBs) between the genomes, with red color showing the connection between LCBs that are in the same orientation and blue color showing the connection between LCBs that are in the opposite orientation. Conserved and highly related regions are deeply colored. The tree shown on the left represents the guide tree of the Mauve alignment.

**Table 1 microorganisms-08-00634-t001:** Differential characteristics between the three novel species and the type strains of phylogenetically related species of the family *Geobacteraceae*.

Characteristic.	1	2	3	4	5	6	7	8	9
Optimal temperature (°C)	30–33	30–33	30–33	25	33	35	30	30–35	30–33
Optimal pH	5.5–6.5	6.0–7.0	5.5–6.5	6.5–7.0	7.0–8.0	6.8	7.0	ca. 6.7	6.0–7.0
G + C content (mol%) *	59.0	59.7	58.4	50.2 ^a^	58.5	54.8	52.8	56.6	61.2
Motility	+	+	+	−	−	+	−	−	+
Electron acceptor usage									
Nitrate	−	−	+	−	−	+	+	+	+
Fumarate	+	+	+	+	−	+	+	−	+
Sulfur	−	−	−	−	−	+	+	−	−
MnO_2_	+	−	+	+	ND	+	−	+	−
Fe(III) citrate	+	+	+	+	ND	+	−	+	−
electron donor usage									
Succinate	+	+	+	+	ND	−	−	−	+
Butanol	−	−	−	−	+	ND	ND	+	−
Ethanol	+	−	+	−	+	−	−	+	+
Propionate	+	+	+	+	−	−	−	+	+
Malate	+	+	+	+	−	ND	−	−	+
Lactate	+	−	+	+	+	−	−	−	+
Methanol	+	+	+	+	−	−	ND	−	+
Pyruvate	+	+	+	+	+	+	−	+	+
Glucose	+	+	+	+	−	−	ND	−	+
Toluene	−	−	−	−	ND	−	ND	+	−

1. *Oryzomonas japonicum* Red96^T^ (data from this study); 2. *Oryzomonas sagensis* Red100^T^ (data from this study); 3. *Oryzomonas ruber* R88^T^ (data from this study); 4. *Geobacter chapellei* DSM 13688^T^ (data from this study); 5. *Pelobacter propionicus* Ott Bd 1^T^ (data from Schink, 1984 [52]); 6. *Geobacter lovleyi* SZ^T^ (data from Sung et al., 2006 [53]); 7. *Geobacter thiogenes* K1^T^ (data from Nevin et al., 2007 [54]); 8. *Geobacter metallireducens* GS-15^T^ (data from Lovley et al., 1993 [3]; Zhou et al., 2014 [6]); 9. *Geomonas oryzae* S43^T^ (data from Xu et al., 2019 [2]). +, Positive; −, negative. ND, no data. * Data from genomic analysis except: ^a^, data from HPLC [55].

**Table 2 microorganisms-08-00634-t002:** General genome features of three novel species: *Oryzomonas japonicum* Red96^T^, *O. sagensis* Red100^T^, and *O. ruber* Red88^T^.

Features	Red96^T^	Red100^T^	Red88^T^
Assembled contigs	30	17	16
Genome length (bp)	3,624,587	3,609,742	3,798,725
N50 length (bp)	336,264	620,468	353,860
Average G + C content (mol%)	59.0	59.7	58.4
Number of predicted ORFs	3363	3322	3523
Number of rRNAs	5	4	3
Number of tRNAs	49	51	50
Number of ncRNAs	3	3	3
Depth of coverage (×)	808	958	271
Accession number	VZQZ00000000	VZRA01000000	SRSD01000000

**Table 3 microorganisms-08-00634-t003:** Genomic similarities (%) of ANI, GGDC, AAI, and POCP between the three strains and other type species in the family *Geobacteraceae*.

Reference Strains ^#^	ANI Value (%)			GGDC Value (%)			AAI Value (%)			POCP (%)		
	Red96^T^	Red100^T^	Red88^T^	Red96^T^	Red100^T^	Red88^T^	Red96^T^	Red100 ^T^	Red88 ^T^	Red96 ^T^	Red100 ^T^	Red88 ^T^
*Oryzomonas japonicum* Red96^T^	100			100			100			100		
*Oryzomonas sagensis* Red100^T^	94.2	100		58.2	100		95.0	100		86.0	100	
*Oryzomonas ruber* Red88^T^	95.6	94.6	100	67.5	61.7	100	96.3	95.2	100	85.0	83.5	100
*Pelobacter propionicus* Ott Bd 1^T^	73.8	74.0	73.9	20.6	21.1	21.1	66.6	66.6	66.4	52.6	51.8	50.8
*Geobacter lovleyi* SZ^T^	70.7	70.6	70.8	19.8	19.9	19.8	63.6	63.6	63.4	56.3	55.4	54.8
*Geobacter thiogenes* ATCC BAA-34^T^	70.3	70.0	70.3	19.2	20.0	19.7	63.7	63.5	63.2	55.8	54.7	53.6
*Geobacter metallireducens* GS-15^T^	70.8	70.9	70.8	20.2	19.9	20.2	62.3	62.4	62	57.8	57.3	56.6
*Geobacter uraniireducens* Rf4^T^	70.7	70.6	70.4	20.2	20.6	20.3	62.8	62.1	62.3	54.3	53.7	53.0
*Geobacter toluenoxydans* JCM 15764^T^	69.7	69.8	69.5	20.0	20.5	19.3	61.1	60.8	60.5	49.4	49.0	47.8
*Geobacter daltonii* FRC-32^T^	69.6	69.5	69.3	20.3	22.2	20.6	61.3	61.0	60.7	54.9	54.5	53.1
*Geobacter sulfurreducens* PCA^T^	70.8	70.8	70.9	20.0	20.5	20.2	62.6	62.7	62.3	60.3	59.7	59.0
*Geobacter anodireducens* SD-1^T^	71.0	71.0	71.1	19.7	20.2	20.0	61.4	61.4	61.3	53.5	53.2	52.5
*Geobacter pickeringii* G13^T^	71.5	71.4	71.5	20.3	20.7	20.6	63.3	63.3	63.4	61.6	61.6	60.4
*Geobacter soli* GSS01^T^	71.0	71.0	71.0	19.8	20.3	20.1	62.4	62.3	62.3	59.4	58.9	58.4
*Geomonas oryzae* S43^T^	70.3	70.5	70.3	19.3	19.4	19.3	60.8	60.7	60.7	57.8	56.9	56.9
*Geomonas edaphica* Red53^T^	70.3	70.3	70.2	19.2	19.1	19.3	60.9	60.9	60.6	58.0	56.7	56.8
*Geomonas ferrireducens* S62^T^	70.3	70.3	70.3	19.3	19.2	19.5	60.9	60.6	60.5	57.4	56.0	55.8
*Geomonas terrae* Red111^T^	69.8	70.0	69.9	19.0	19.3	19.2	60.5	60.6	60.3	58.1	56.6	56.7
*Geomonas bemidjiensis* Bem^T^	70.2	70.3	70.2	19.8	19.8	19.9	60.6	60.6	60.5	56.8	56.1	56.1
*Geomonas bremensis* R1	70.4	70.2	70.3	19.9	19.9	19.9	60.4	60.4	60.1	56.4	55.7	55.5

^#^ Genomes of all reference strains were retrieved from the NCBI database with the accession numbers and genome size listed in Appendix A.

**Table 4 microorganisms-08-00634-t004:** Fatty acid compositions of the three novel strains and the reference strain *Geobacter chapellei* DSM 13688^T^.

Fatty Acid	1	2	3	4
C_12:0_	-	-	0.1	1.9
iso-C_13:0_	0.2	0.3	0.3	2.6
C_14:0_	5.5	9.4	9.9	4.5
iso-C_15:0_	52.1	40.4	33.5	19.1
a-C_15:0_	0.7	0.5	0.5	0.8
C_15:0_	0.4	0.6	0.4	0.4
C_15:1_ B/F *	1.4	1.3	1.3	0.4
iso-C_16:0_	1.7	0.8	0.8	1.3
C_16:0_	7.4	10.8	13.8	20.1
C_16:1_ B/F *	1.9	0.5	0.4	1.9
C_16:1_ *ω7c*	19.2	22.2	26.0	32.9
C_16:1_ *ω5*c	0.7	0.6	1.0	2.1
C_16:0_ 10-methyl	0.9	0.6	0.8	ND
C_16:0_ 3-OH	2.2	7.0	7.4	7.3
C_17:1_ B/F *	1.6	0.6	0.5	0.3
iso-C_17:0_	1.5	0.9	0.8	0.9
a-C_17:0_	0.1	0.1	0.1	0.2
C_18:0_	0.3	0.6	0.6	1.1
C_18:1_ *ω*7c	1.5	1.1	1.0	0.4

1. *Oryzomonas japonicum* Red96^T^; 2. *Oryzomonas sagensis* Red100^T^; 3. *Oryzomonas ruber* R88^T^; 4. *Geobacter chapellei* DSM 13688^T^. All data listed in this table derived from this study, only those accounting for 0.2% or more of the total in one or more of the strains are given. -, trace quantities (<0.1%). ND, not detected. * The double bond position could not be identified (except for C_15:1_
*ω5*c, C_16:1_
*ω5*c, C_16:1_
*ω7*c, and C_17:1_
*ω7*c).

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
