# Peer review of "Description of Three Novel Members in the Family Geobacteraceae, Oryzomonas japonicum gen. nov., sp. nov., Oryzomonas sagensis sp. nov., and Oryzomonas ruber sp. nov."

_microorganisms, 2020, doi:10.3390/microorganisms8050634_

Round 1

Reviewer 1 Report

I found the manuscript: “Oryzomonas japonicum gen. nov., sp. nov., Oryzomonas niigatensis sp. nov., and Oryzomonas ruber sp. nov., three novel members of the family Geobacteraceae isolated from paddy soil and pond sediment” to be very interesting and of high quality. The authors presented an overall well-organized experiment, the results are well-presented and constitute a nice data set regarding the three new novel species. This study is of the same quality as the previous work of the authors (Xu et al. 2019), offering new information about a novel genus in the Geobacteraceae family, and amplifying our limited knowledge about the Geobacteraceae species participating in the biogeochemical processes in paddy soils and sediment.

My only suggestion to the reviewers would be to include in their introduction more information about the importance of these bacteria in sediments. Even though one of the new species was isolated from pond sediment, the authors did not justify why they chose to do that in their introduction, making it unclear why they included sediment analysis in their work.

Author Response

Response to Reviewer 1 Comments

I found the manuscript: “Oryzomonas japonicum gen. nov., sp. nov., Oryzomonas niigatensis sp. nov., and Oryzomonas ruber sp. nov., three novel members of the family Geobacteraceae isolated from paddy soil and pond sediment” to be very interesting and of high quality. The authors presented an overall well-organized experiment, the results are well-presented and constitute a nice data set regarding the three new novel species. This study is of the same quality as the previous work of the authors (Xu et al. 2019), offering new information about a novel genus in the Geobacteraceae family, and amplifying our limited knowledge about the Geobacteraceae species participating in the biogeochemical processes in paddy soils and sediment.

My only suggestion to the reviewers would be to include in their introduction more information about the importance of these bacteria in sediments. Even though one of the new species was isolated from pond sediment, the authors did not justify why they chose to do that in their introduction, making it unclear why they included sediment analysis in their work.

Thank you very much for the positive feedback and valuable comment to improve our manuscript.

Descriptions of motivation for isolation from freshwater sediments have been added in introduction part of revised manuscript (Line 88-94).

Reviewer 2 Report

The manuscript reports on the isolation of three novel species of the family Geobacteraceae from wetlands in Japan.

Detection of these new Fe-reducting isolates is novel and of value for soil and environmental research. The manuscript is well written and can be accepted for publication after a round of revisions.

I have a few suggestions.

Please consider revising the title, I believe the second part of the title should come first.

If possible, please include a geographical map of the sampling sites.

Line 38: please revise to "Gram-negative"

Line 39: is there a picture for the agar plates?

Please consider highlighting the importance of the novel isolates in terms of their potentials for studies of biogeochemical processes.

Author Response

Response to Reviewer 2 Comments

The manuscript reports on the isolation of three novel species of the family Geobacteraceae from wetlands in Japan.

Detection of these new Fe-reducting isolates is novel and of value for soil and environmental research. The manuscript is well written and can be accepted for publication after a round of revisions.

Thank you very much for the positive feedback and valuable comment to improve our manuscript.

I have a few suggestions.

Please consider revising the title, I believe the second part of the title should come first.

Thanks for the suggestion. We have changed the title as “Description of three novel members in the family Geobacteraceae, Oryzomonas japonicum gen. nov., sp. nov., Oryzomonas niigatensis sp. nov., and Oryzomonas ruber sp. nov.” (Line 2-5)

If possible, please include a geographical map of the sampling sites.

Thanks for the suggestion. We have added the geographical map accordingly in the supplementary document as the Figure S1.

Line 38: please revise to "Gram-negative"

Thanks for the suggestion. We have modified accordingly. (Line 37, 216, 481)

Line 39: is there a picture for the agar plates?

Thanks for the suggestion. We have added the pictures of agar plates into Figure 1 in the revised manuscript.

Please consider highlighting the importance of the novel isolates in terms of their potentials for studies of biogeochemical processes.

Thanks for the suggestion. We have added one paragraph in part “Conclusion” to highlight the importance of these three isolates in the potentials for biogeochemical studies. (Line 463-470)

Round 2

Reviewer 2 Report

The authors have responded satisfactorily to my previous comments and have made the necessary revisions. I think the manuscript can be accepted for publication.

Author Response

The authors have responded satisfactorily to my previous comments and have made the necessary revisions. I think the manuscript can be accepted for publication.

Thank you very much for the positive feedback.